# Dismantling the Component-Specific Effects of Yogic Breathing: Feasibility of a Fully Remote Three-Arm RCT with Virtual Laboratory Visits and Wearable Physiology

**DOI:** 10.3390/ijerph20043180

**Published:** 2023-02-11

**Authors:** Yan Ma, Huan Yang, Michael Vazquez, Olivia Buraks, Monika Haack, Janet M. Mullington, Michael R. Goldstein

**Affiliations:** 1Osher Center for Integrative Medicine, Division of Preventive Medicine, Brigham and Women’s Hospital, Harvard Medical School, Boston, MA 02115, USA; 2Department of Neurology, Beth Israel Deaconess Medical Center, Harvard Medical School, Boston, MA 02115, USA; 3Department of Internal Medicine, Brookdale Hospital Medical Center, Brooklyn, NY 11212, USA

**Keywords:** slow-paced breathing, mindfulness, mind-body intervention, remote, sleep, wearable physiology, heart rate, stress

## Abstract

Despite the growing research base examining the benefits and physiological mechanisms of slow-paced breathing (SPB), mindfulness (M), and their combination (as yogic breathing, SPB + M), no studies have directly compared these in a ”dismantling” framework. To address this gap, we conducted a fully remote three-armed feasibility study with wearable devices and video-based laboratory visits. Eighteen healthy participants (age 18–30 years, 12 female) were randomized to one of three 8-week interventions: slow-paced breathing (SPB, *N* = 5), mindfulness (M, *N* = 6), or yogic breathing (SPB + M, *N* = 7). The participants began a 24-h heart rate recording with a chest-worn device prior to the first virtual laboratory visit, consisting of a 60-min intervention-specific training with guided practice and experimental stress induction using a Stroop test. The participants were then instructed to repeat their assigned intervention practice daily with a guided audio, while concurrently recording their heart rate data and completing a detailed practice log. The feasibility was determined using the rates of overall study completion (100%), daily practice adherence (73%), and the rate of fully analyzable data from virtual laboratory visits (92%). These results demonstrate feasibility for conducting larger trial studies with a similar fully remote framework, enhancing the ecological validity and sample size that could be possible with such research designs.

## 1. Introduction

Slow-paced breathing (SPB), mindfulness (M), and their combination (SPB + M) in the form of yoga, yogic breathing, Tai Chi, Qigong, and other practices are increasingly being offered in clinical and wellness settings [1,2]. Correspondingly, there has been accumulating interest in the potential benefits of these practices on health and wellness, including stress [3,4,5,6,7], sleep [8,9,10,11,12], and autonomic function [13,14], among other domains. Although the scientific community continues to build an understanding of the autonomic mechanisms that might be unique to SPB [15,16], M [17,18], and their interaction [7], no studies to our knowledge have attempted to directly compare these three forms of intervention in a controlled fashion that enables a “dismantling” framework of interpretation.

While some individuals are attracted to mindfulness with its growing popularity, some individuals are reluctant due to any number of reasons ranging from a perceived stigma, known others who did not report any benefit, or a simple preference to engage in a more physically engaging practice. Individual preferences can greatly influence which intervention someone chooses and/or adheres to [1,19]. From an evidence-based healthcare perspective, it behooves the mind-body medicine community to better understand these interventions in terms of their active components. A better understanding of the effects of SPB directly compared to M and SPB + M may bolster the confidence in recommending SPB as a stand-alone approach (and vice-versa for M, although some form of intentional breathing is commonly integrated in M). Conversely, if a significant benefit is observed for the interactive effects of SPB + M beyond SPB or M alone, this result could indicate a need for further examination of the unique additive effects.

Understanding the unique effects of SPB and M versus their combination also has implications for understanding their mechanisms of action [7,20]. In contrast with cognitive approaches to stress management and autonomic regulation that are relatively more control-based (e.g., stress reappraisal and “positive thinking”) [21], mindfulness practices take an acceptance-based approach. A definition commonly used in mindfulness research is “the awareness that arises from paying attention, on purpose, in the present moment, and non-judgmentally” (Kabat-Zinn [22]). Mindfulness also differs from SPB in that the physiological mechanisms more likely involve top-down (i.e., cortical) activation, whereas SPB engages bottom-up mechanisms (i.e., the direct innervation of the peripheral nervous system via the afferent receptors) [20].

To address these clinical research gaps, we tested the feasibility of a three-arm intervention trial, involving a 20-min daily practice for 8 weeks with wearable devices for the measurements before and after the intervention. We chose to first focus on a healthy young adult population to then inform the translation to hypertension, insomnia, and other clinical groups. Telehealth and other technology-based approaches can improve accessibility to healthcare and decrease the burden associated with scheduling and travel, costs, long wait lists, or a lack of trained providers. Furthermore, given the increasing number of studies with remotely delivered mindfulness-based interventions [23,24,25,26,27,28,29,30], and the pragmatic limitation that the study was conducted during the COVID-19 pandemic, we designed this study to be fully remote. The existing studies have provided evidence that the technology in wearable heart rate monitors is relatively mature, and heart rate variability (HRV) has been one of the most commonly used physiological measurements for both breathing exercises or mindfulness-based interventions [7,31,32,33,34]. The prospective feasibility for an adequately powered follow-on study was determined using the rates of attendance, adherence, and completion as the primary outcome measures. We also qualitatively explored the patterns of intervention-specific changes (HRV-derived breathing rate and mindfulness self-ratings) as the preliminary indicators of promise for an intervention dismantling framework. However, consistent with the guidelines from the NCCIH [35] as well as Orsmond & Cohn [36] for feasibility studies, we limit our reporting to descriptive statistics and data visuals specific to intervention feasibility-related data, without inferential statistics.

## 2. Materials and Methods

We employed a fully remote study design of a three-arm randomized clinical trial (RCT, registered at clinicaltrials.gov, NCT04866901). The Institutional Review Board at Beth Israel Deaconess Medical Center approved this study. The recruitment started in May 2021, and the data collection was completed in September 2021, during the COVID-19 pandemic. 

### 2.1. Eligibility Criteria and Randomized Trial Design

In this study, we recruited healthy young adult participants aged 18–35 years. We excluded participants with (1) an active infection/disease, (2) a current untreated mental or physical health condition deemed likely to interfere with their ability to complete the study procedures (determined by study staff consensus), (3) current use of medications with known effects on stress physiology, including antidepressants (SSRI, SNSI, NDRI, atypical, TCA, MAOI), antipsychotics, benzodiazepines, non-benzodiazepine receptor agonists, melatonin and melatonin receptor agonists, orexin/hypocretin receptor antagonists, barbiturates, mood stabilizers, anticonvulsants, anticholinergics, first generation antihistamines, and stimulants including NRI, antihypertensives, opioids, or systemic corticosteroids, (4) current pregnancy, and (5) moderate/substantial prior meditation, yoga, or other mind-body practice reported as a self-rating of 5 or higher on a scale of 0–10 asking “How experienced are you with meditation, yoga, or other mind-body interventions?”. 

### 2.2. Recruitment and Consent

Due to the remote nature of the study design, we recruited the participants through ResearchMatch.org, a national electronic, web-based recruitment tool that was created through the Clinical and Translational Science Awards Consortium in 2009 and is maintained at Vanderbilt University as an IRB-approved data repository. Internet postings and local flyers were also used. A preliminary screening was conducted via telephone and/or email when someone with an interest in participating contacted the research office. Potentially eligible individuals were then invited to a virtual meeting with our study team to go through the informed consent process and complete questionnaires designed to assist in determining eligibility. The initial screening and informed consent procedures were conducted remotely via an IRB-approved videoconferencing platform (Starleaf). Once enrolled, a research kit consisting of wearable devices and instructions were mailed to each participant. The scheme of the study design and a timeline is shown in Figure 1.

### 2.3. Randomization and Interventions

After granting informed consent, the participants were randomized to one of the intervention groups: slow-paced breathing (SPB), mindfulness (M), or yogic breathing (SPB + M). All three conditions involved a 20-min structured introduction and detailed instructions unique to their assigned intervention, followed by a 5-min opportunity for an initial practice and questions, a 20-min formal practice opportunity with physiological recordings, and finally, corrective feedback and the opportunity to address questions or challenges regarding the practice (60-min total). The participants were provided with a 20-min audio recording to guide their home practice, comparable in format across the three conditions. 

Slow-paced Breathing (SPB): The participants were provided with a brief overview of the science of breathing and the benefits for autonomic regulation, then were guided via audio recording to breathe at a constant rate of 6 breaths/min (5 s inhale, 5 s exhale). The guided audio emphasized the importance of following the specific rhythm of breathing, without regard to thoughts or inner experience. A soft but firm tone of voice was employed to minimize the likelihood of relaxing effects, while maintaining similarity to the tone of voice used in the other conditions.

Mindfulness (M): The procedures were broadly based on a prior study by Berghoff et al. [37], providing a brief history of mindfulness practices, definitions, instructions for practice, common challenges, and recommendations. An audio recording then guided the mindfulness practice. Specific to this study, in order to further distinguish the three conditions, the guided audio recording emphasized the importance of attending to the quality of the experience while not changing or attending to breathing patterns. 

Yogic breathing (SPB + M): The procedures for the other two conditions were integrated with the aim of eliciting the attention to the same breathing instructions used for SPB, while also observing the quality of the experience during the practice, as conducted for M. The guided audio duration and the frequency of instructions was similar in the SPB + M condition as the other two conditions.

### 2.4. Virtual Lab Visits

A virtual lab visit (approximately 90-min) consisted of the following. First, a study staff member greeted the participant on the video call, confirmed their current location in case of any emergencies, answered any urgent questions, and provided an overview of the session. Next, the staff member ensured the correct placement of the devices by asking the participant to demonstrate this by video camera, and made any necessary corrections (e.g., tightening the bands if they appeared loose). Then, the staff member instructed the participant to rest quietly for a 10-min baseline period. After this resting period, the study interventionist (M.R.G.) joined the video call to provide the 60-min training for their assigned intervention (see below for details). Next, the interventionist left the call and the study staff member managing the visit returned to guide the participant through the online Stroop test. Finally, the staff member answered any remaining questions and completed the call. Timestamps for each period of interest (resting baseline, intervention practice, and stress task) were tracked by the study staff to align with the ambulatory recording devices for the subsequent analysis.

Stroop stress task: To also test the feasibility of probing the effects of the interventions on a simulated condition of daily stress, an online Stroop test was administered during each of the virtual laboratory visits, following the same procedure as Chin and Kales [38,39] (http://cognitivefun.net (accessed on 5 February 2023)). 

### 2.5. Baseline Questionnaire-Based Measures

To provide additional baseline characteristics relevant to feasibility and the recruitment of target populations for potential follow-on studies, we collected and reported a select set of commonly used questionnaires, including the Perceived Stress Scale (PSS-14) [40,41], Pittsburgh Sleep Quality Index (PSQI) [42,43,44,45,46], Mood and Anxiety Symptom Questionnaire (MASQ) [47,48,49], and Big Five Inventory of personality (BFI) [50].

### 2.6. Daily Logs

Daily sleep logs were completed electronically (via REDCap accounts) for a 2-week period before and after the 8-week training period to examine the day-to-day patterns of sleep, stress, and mood, following a format commonly used in our laboratory [51]. The daily practice adherence during the 8-week intervention period was tracked with a daily log format similar to the sleep logs, as well as the completion of heart rate recordings, completed in the Elite HRV mobile app. Each daily log entry was identifiable with a unique timestamp of completion and a participant code to enable detailed organization for analysis.

### 2.7. Ambulatory Heart Rate and Heart Rate Variability

The continuous heart rate data were recorded using a chest-worn bipolar electrode transmitter belt (Polar H9, Finland), the successor model for the one used in the two key reference studies with stress induction tasks described above [38,52]. The participants were asked to wear the heart rate monitor chest strap for 24 h overlapping with their pre- and post-intervention laboratory visits, as well as during their daily practice. The heart rate data and time stamps were extracted for analyses including heart rate variability (HRV). HRV is the most commonly used measurement for autonomic regulation, and it typically includes time and frequency domain analyses. HRV power spectrum measurements (frequency domain) are often used to reflect the respiration rate derived from heart rate data. Because HRV is not a primary outcome, and this paper is only examining feasibility, here we only report a respiration-related frequency domain analysis to test whether the participants followed paced breathing during the virtual lab visits and to explore the intervention-related changes in breathing rates at a resting state. Paced breathing at 6 breaths per minute (bpm) generates a peak at 0.1 Hz on the spectrum, and breathing at 15 bpm, for example, will result in a peak at 0.25 Hz. We employed a spectral analysis from HRV and incorporated it with customized scripts in Matlab (MathWorks, Natick, MA, USA) for visual inspections of breathing and adherence. 

### 2.8. Outcome Measurements

Our primary outcomes for this feasibility study included the screening-to-randomization ratio (the percentage of all eligible patients who were randomized), the overall completion rate (the completion of all the procedures including the intervention and assessments; the pre-determined feasibility threshold was ≥70%) [37,53], the overall daily practice adherence rate (based on the practice log; feasibility threshold: ≥50%) [37,54], the completion rate of daily sleep logs (the averaged completion rates of each participant; feasibility threshold: ≥70%) [55], and the rate of analyzable data from the virtual lab visits (feasibility threshold: ≥70%) [53,56]. In addition to these, we also examined the patterns of the HRV-derived breathing rate and self-rated mindfulness as a preliminary, qualitative approach to explore the feasibility of distinguishing component-specific patterns of mindfulness and breathing interventions, including whether we could obtain a distinguishable baseline from practice and VLV recording blocks, whether we could distinguish varying levels of adherence based on the breathing rate estimate from HR data, whether breathing groups follow the instructed breathing rhythm, whether the groups were distinguishable in terms of HRV spectra, and whether only the mindfulness groups show an increase in mindfulness.

Additional procedures and measures that were collected but not reported here include a variety of questionnaires, momentary stress ratings during the laboratory visits, and qualitative feedback from the debrief interview upon study completion. 

### 2.9. Statistical Analyses

Per the guidelines for feasibility studies set forth by the NCCIH and others [57], we did not power this study for testing the intervention efficacy or group-wise differences, thus we only report the descriptive statistics and data visuals here. The planned target of 15 participants was based on the study needs and pragmatics (including time and budget considerations). The continuous data are reported in text and tables with means and standard deviations for normally distributed data and with medians and interquartile ranges for skewed data. Discrete data were reported as absolute numbers and percentages. 

## 3. Results

### 3.1. Feasibility of the Remote RCT

See Figure 2 for the CONSORT study flow diagram. Our recruitment started in May 2021, and the data collection was completed in September 2021, which was in the midst of the COVID-19 pandemic. During the 6 months we screened 91 individuals with an interest in participating, and we were able to recruit 19 eligible participants for this study. Of these 19 participants, one withdrew prior to randomization, and 18 were randomized to one of three 8-week interventions: slow-paced breathing (SPB, *N* = 5), mindfulness (M, *N* = 6), or yogic breathing (SPB + M, *N* = 7). Two participants joined the study together and were cluster-randomized to SPB + M, thus resulting in the varying sample sizes. See Table 1 for the demographic characteristics. Of the 18 participants, 10 were full-time students, 7 were working professionals, and 1 was unemployed at the time of participation. The resulting screening-to-randomization ratio was 18/91 (20%). All 18 randomized participants completed all the procedures including the intervention and assessment, and the questionnaires for self-esteem, satisfaction with life, social connectedness, emotion regulation, desirability of control, psychological flexibility, and circadian patterns (not reported here); therefore, the overall completion rate was 100%. The completion rate of the remotely conducted laboratory visits was 88.9%, due to technical issues with the Stroop test on four occasions. The overall daily practice adherence rate was 72.8%, with the SPB and SPB + M groups relatively higher (77.1% and 76.5%, respectively) and the M group lowest at 64.9%. We noted a progressive decline in these group-averaged rates over the 8 weeks (Table 2), with the exception of one participant who demonstrated an opposite pattern by increasing in adherence as the intervention period progressed. The daily sleep logs were successfully collected with REDCap using automatically scheduled reminders, and then inspected with an automated procedure for common errors or potential concerns over data quality. On average, the completion rate of the daily sleep logs was 97.6% pre intervention and 91.7% post intervention. Again, the M group showed a relatively lower completion rate of 83.3% post intervention, compared to the other two groups (100% and 92.9%). From the HR data collected through the wearable devices, we were able to visually examine the adherence of participants following paced breathing compared to the other recording blocks of the virtual lab visit (Appendix A), with 15 of the 18 participants recording complete, analyzable data during the virtual lab visit pre intervention and all 18 post intervention (91.7% overall).

### 3.2. Feasibility of Wearable Devices to Detect Intervention-Specific Breathing Patterns

Using the Polar H9 wearable device, we were able to collect continuous heart rate data at all the designated sessions. In our supplementary figures, we show representative examples of individual participant data from the virtual laboratory visit (Appendix A), individual participant data for one week of daily practice logs and HR recordings (Appendix A), a comparison between the SPB participants with high versus low adherence to the guided breathing rhythm throughout the study (Appendix A), the varying timecourse patterns of adherence within an individual SPB participant across the practice days (Appendix A), and the group-averaged timecourse of HRV spectral power during the practice sessions (Appendix A). In summary, HRV-derived breathing patterns were able to clearly differentiate mindfulness versus the guided breathing practices, as well as the distinguishably varying levels of adherence to the guided breathing instruction. For example, the peaks at 0.1 Hz (equivalent of 6 breaths/min) were easily identifiable in the groups that practiced slow paced breathing (i.e., the SPB and SPB + M groups), but not the M group (Figure 3). When examining the individual data for the HRV-derived breathing rates, we see that the 6 breaths/min instruction was followed in all but one instance for the SPB and SPB + M group (Figure 4, left panel). Conversely, the breathing rates for the M participants varied widely, from around 4 breaths/min to 17 breaths/min. When examining the HRV spectra from the 5-min resting baseline recordings collected during the daily practice sessions, we did not observe an obvious difference between the groups or a progressive change across the 8-week intervention period (Appendix A). In other words, the SPB and SPB + M groups did not appear to integrate the 6 breaths/min rhythm from the guided practice into their resting breathing patterns any more than the naturally evident 0.1 Hz peak observed for the M participants, reflective of the spontaneous Meyer wave rhythm.

### 3.3. Mindfulness

The two mindfulness groups, M and SPB + M, demonstrated relatively more consistent patterns of increased mindfulness self-ratings from pre to post training compared to SPB, as assessed by both the Langer Mindfulness Scale (SPB: 40%, M: 67%, SPB + M: 71%) and the Five Facet Mindfulness Questionnaire (SPB: 20%, M: 83%, SPB + M: 43%), shown in Table 3 and Figure 3.

## 4. Discussion

This study successfully demonstrated the feasibility of a three-arm intervention trial, involving a 20-min daily practice for 8 weeks, conducted fully remotely with wearable devices and a virtual laboratory visit with a stress induction task. Our research team screened 91 individuals who showed an interest in participating, and then successfully enrolled and randomized 18 healthy young adults (5 in the SPB group, 6 in the M group, and 7 in the SPB + M group) from multiple states in the U.S. over a 5-month period, at a rate of about one participant per week. With a 20% recruitment-to-randomization rate, we then achieved our target rates of overall study completion (100%), daily practice adherence (73%), daily sleep log completion (98%), and rate of analyzable heart rate data from virtual lab visits (92%), indicating that a larger follow-on study with these procedures is feasible. Furthermore, while this feasibility study was not sufficiently powered for statistical analysis, we observed intervention-specific patterns of breathing and mindfulness consistent with a “dismantling” framework, whereby only the SPB and SPB + M participants demonstrated spectral HRV patterns reflecting the 6-bpm guidance, and only M and SPB + M yielded group-wise increases in the mindfulness ratings from pre to post intervention. In addition to the measures reported here, we collected other questionnaires for the exploratory aims, including scales for self-esteem, satisfaction with life, social connectedness, emotion regulation, desirability of control, psychological flexibility, and circadian patterns, as well as performance metrics from the Stroop test. The data collection rates for these secondary measures were similarly high.

The primary aim of the study was to test the feasibility of dismantling the component-specific effects of yogic breathing (mindfulness, breathing, and their combination) via a fully remote three-arm RCT. Therefore, we limited our focus to the feasibility-related data rather than the “outcomes” and we reported the results with descriptive statistics and data visuals but not the inferential statistics, per the recommended guidelines [35]. For this particular study design, the ”outcomes” would include any clinically relevant changes beyond breathing or mindfulness, such as changes in blood pressure or the questionnaire metrics in Table 1 and others described in the paragraph above. Conversely, in this study, the HRV spectra and HRV-derived breathing rate, along with the LMS and FFMQ scores for mindfulness, are extended feasibility measures beyond the adherence rates that still reflect the feasibility of conducting these interventions in the proposed dismantling framework, akin to a manipulation check.

Recognizing that many mindfulness and yoga interventions are much longer in duration, such as Mindfulness-Based Stress Reduction (MBSR) that entails 2.5-h weekly group sessions for 8 weeks plus a full-day retreat and 45-min daily formal practice [58], we opted to use this abbreviated version for the following reasons: (a) brief and even infrequently practiced interventions have produced similar benefits as longer versions [59]; (b) lengthy instruction and home practice expectations can negatively impact participation [60]; (c) Berghoff et al. [37] found similar effects on self-report measures for 2-week 10-min and 20-min practices with similar adherence between the two conditions; (d) the actual practice time observed in a seminal MBSR study [58] was <17-min on average, despite the instructed 45-min, and the practice time was not correlated with the immune response that showed a significant group-wise effect; and (e) most paced-breathing interventions use daily practice durations of <20-min [61], rendering the 20-min duration to be a compromise for an equivalent instructed practice across all conditions. Nevertheless, we observed slightly (numerically) lower rates of daily practice adherence for the M-only intervention group, possibly due to the relatively lower amount of behavioral engagement inherent to mindfulness alone compared to breathing-based interventions or mindfulness combined with breathing.

In this feasibility study, we focused on a healthy young adult population, and used technology-based approaches to improve accessibility and decrease the participants’ burden. Compared to the M and SPB + M groups, the SPB group had higher MASQ scores for in three of the scales: distress, anxious arousal, and total mood disturbance. However, no differences were observed for the feasibility outcomes, suggesting that the study is potentially feasible for people with higher levels of stress, depression, and/or anxiety. Correspondingly, this type of study design may have a promising application for hypertension, insomnia, and other clinical groups. Studies with remotely delivered mindfulness-based interventions have been increasing [23], especially during the COVID-19 pandemic. An increasing number of studies have aimed to investigate the feasibility of remotely delivered mindfulness-based interventions involving various populations, for example, perinatal women [26,62], elementary school students [63], adolescents [24], advanced cancer patients and their informal caregivers [64], adult cigarette smokers [65], and medical students/professionals [25,62]. Collectively, these studies have reported improvements in resilience, stress management, depression, anxiety, emotional wellbeing, sleep, and mindfulness. However, very few of these prior studies included physiological data collection, and when collected, the physiological data has been limited (e.g., single data points rather than capturing the richness of a timecourse of physiological changes). We hope that our inclusion of wearable heart rate monitors with detailed data visuals of the beat-to-beat timecourse data will encourage future studies with remote designs to also include such objective/physiological measures and a timecourse analysis.

As highlighted above, the three groups were qualitatively distinguishable in terms of the HRV spectra during practice (Figure 3), with identical peaks at 0.1 Hz in the SPB and SPB + M groups. During the daily resting baseline across the 8-week training period, we observed a slight progressive increase in higher-frequency power (>0.15 Hz) in all three groups by comparing the four 2-week intervals, with the SPB group being the most obvious, followed by the M group, and the SPB + M group showing the least increase (Appendix A). However, there was not a progressive shift or visually discernable difference between the groups at 0.1 Hz during the daily resting baseline recordings. 

In this fully remote trial, we encountered a few sporadic technical issues. Because we used a freely available, web browser-based version of the Stroop test, there were two occasions pre intervention and two occasions post intervention (11%) when the server was unavailable to successfully complete the task. In all cases, we used a backup version of the test to approximate the stress induction aspect, although we were not able to obtain comparable performance statistics. Consistent with a common drawback of telehealth, there were also periodic connectivity issues with the video platform, in which case we used telephone calls to maintain contact with the participants to complete as much of the virtual lab visit and wearable data collection as possible. As described above, there were three occasions when we were not able to collect the analyzable heart rate data; in one case, there were connectivity issues for the participant, and in the other two cases, there was a misunderstanding of the recording procedure during the intervention practice recording, which was able to be immediately remedied for all subsequent recordings. Overall, we were encouraged by the relatively few issues that arose and the success of troubleshooting efforts.

A few additional limitations are important to note. Due to local IRB regulations, we were unable to advertise on social media or through university posting boards, and instead, we used ResearchMatch.org as the sole advertising modality. While we garnered substantial participant interest through this resource, our screening-to-randomization ratio was low (20%) and may have been improved had we been able to use more targeted, local advertisement methods. Second, the relatively younger healthy adult population in this study showed a high adherence and reported a low level of difficulty in following the technology-based tests and measures, which may overestimate the anticipated feasibility for older adults or other populations when considering the ease-of-access related to videoconferencing and wearable technology. Future studies are encouraged to test the feasibility of such a design in other populations, especially using a qualitative or mixed methods analysis. It is also notable that two of the M participants breathed at rates near 6 bpm (0.1 Hz), presumably corresponding to an effortful slowing of the breathing, despite no specific instruction to breathe at that rate. The variations in breathing rates across these interventions create a challenge for conducting an HRV-based analysis of the vagal tone or parasympathetic activation per conventional approaches using a cutoff of 0.12–0.15 Hz. However, one possible solution that future studies could implement is a recording block of paced breathing, for example at 15 bpm (0.25 Hz), for all participants at the end of the final laboratory visit, in order to obtain the equivalent breathing conditions while not inadvertently influencing the breathing-related instructions for each intervention arm.

## 5. Conclusions

The feasibility benchmarks met our preset criteria for success, thus this study indicates that a fully remote randomized clinical trial comparing the separate and combined effects of SPB and M is feasible, enhancing the ecological validity and sample size that could be possible with such research designs.

## Figures and Tables

**Figure 1 ijerph-20-03180-f001:**
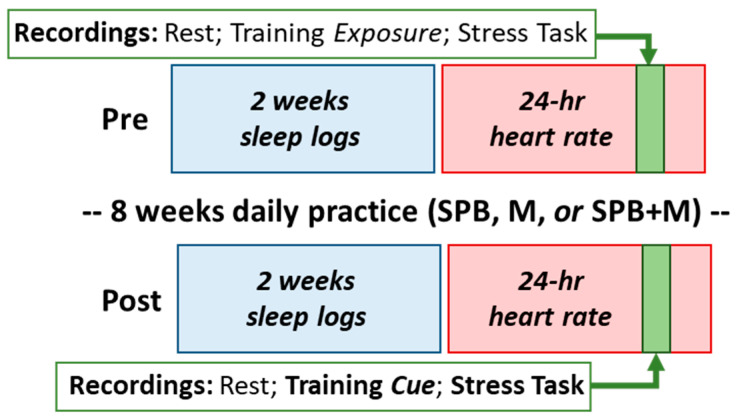
Study design. SPB, slow-paced breathing; M, mindfulness; SPB + M, yogic breathing.

**Figure 2 ijerph-20-03180-f002:**
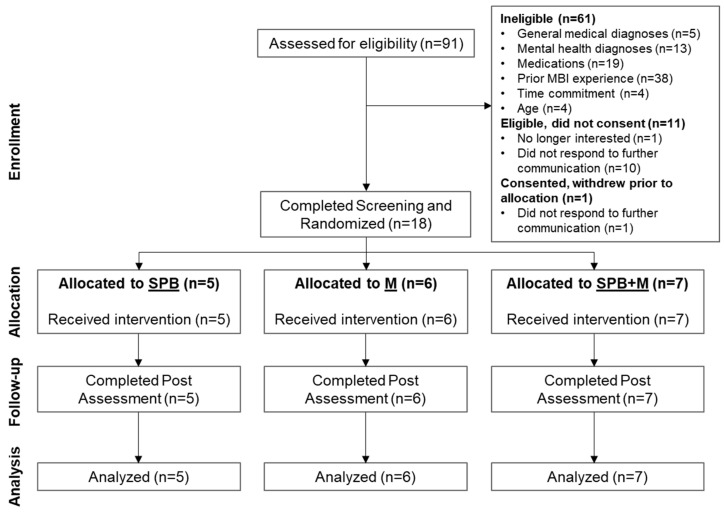
Study flow of the RCT.

**Figure 3 ijerph-20-03180-f003:**
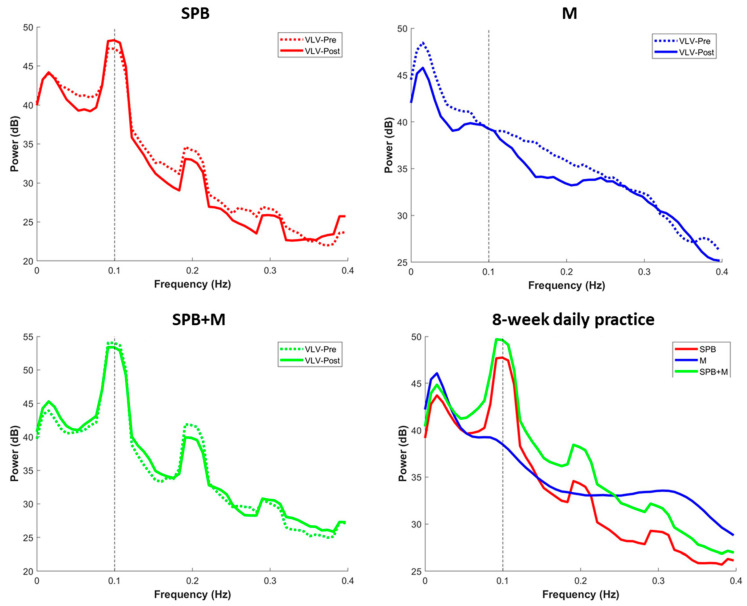
Heart rate variability (HRV) spectral power during the practice sessions at the virtual lab visits (VLV) before and after the intervention (top panels and lower left), and during the daily home practice sessions (lower right). For all figures, the data were averaged across all the available recordings for each participant and then averaged for the group.

**Figure 4 ijerph-20-03180-f004:**
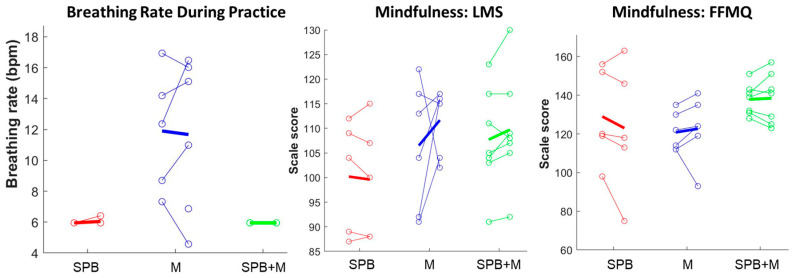
Comparative effects of the interventions on the HRV-derived breathing rate and self-reported mindfulness, plotted as the individual change from pre to post intervention (open circles and connected lines) and averaged for each group (thick lines). Although two M participants breathed at an average rate below 8 bpm, only SPB and SPB + M breathed at a rate centered at 6 bpm (0.1 Hz). Conversely, only the two mindfulness groups, M and SPB + M, showed a group-wise increase in the self-reported mindfulness scales. LMS, Langer Mindfulness Scale; FFMQ, Five Facet Mindfulness Questionnaire.

**Table 1 ijerph-20-03180-t001:** Demographic and baseline characteristics.

	All	SPB	M	SPB + M
Gender (F/M)	12/6	4/1	4/2	4/3
Age (M yrs, SD)	24.2 (2.9)	24.6 (2.1)	23.7 (3.7)	24.3 (3.1)
Ethnicity				
Hispanic	2	1	1	0
Non-Hispanic	16	4	5	7
Race				
Black or African American	0	0	0	0
Asian American	5	1	1	3
American Indian or Alaska Native	0	0	0	0
White	11	3	4	4
Other specified	1	1	0	0
Multi-racial	1	0	1	0
Baseline questionnaires				
Perceived Stress Scale (PSS-14)	14.9 (5.1)	18.6 (5.1)	14.7 (5.5)	12.4 (3.7)
Pittsburgh Sleep Quality Index (PSQI)	4.2 (2.2)	4 (3.6)	4.2 (0.8)	4.4 (2.2)
Mood and Anxiety Symptom Questionnaire (MASQ)				
General Distress	16.2 (7.2)	23.6 (8.6)	14.8 (4.5)	12 (3.6)
Anxious Arousal	13.6 (5.2)	18.8 (7.6)	12 (2.3)	11.3 (1.6)
Anhedonic Depression	19.9 (5.4)	22.4 (7.9)	18.7 (4.8)	19.1 (3.8)
Total mood disturbance	49.7 (15.3)	64.8 (21.5)	45.5 (8.5)	42.4 (4.6)
Big Five Inventory of personality (BFI)				
Extraversion	27.3 (5.7)	28.4 (3.6)	26.5 (6.2)	27.1 (7.1)
Agreeableness	35.7 (4.5)	36.2 (3.6)	33 (4.4)	37.6 (4.6)
Conscientiousness	35.1 (5.7)	35 (6.4)	33.2 (6.1)	36.9 (5.1)
Neuroticism	20.7 (5.2)	21.8 (5.4)	22.7 (3.7)	18.3 (5.7)
Openness	32.3 (4.9)	33.4 (5.2)	32.2 (5.4)	31.6 (5)

**Table 2 ijerph-20-03180-t002:** Completion rates of digital daily sleep logs and practice logs.

	All	SPB	M	SPB + M
Daily sleep logs (%)				
Pre	96.4 (8.9)	92.9 (16)	96.4 (6)	99 (2.7)
Post	91.7 (17.4)	100 (0)	83.3 (28.1)	92.9 (8.2)
Daily practice logs (%)				
Weeks 1–2	80.6 (24.5)	85.7 (16.8)	76.2 (27.4)	80.6 (29.1)
Weeks 3–4	73.4 (23.1)	75.7 (21.8)	64.3 (32.3)	79.6 (13.9)
Weeks 5–6	67.5 (26.4)	68.6 (35.2)	57.1 (28.9)	75.5 (16.4)
Weeks 7–8	64.7 (31.6)	72.9 (29.2)	57.1 (41.4)	65.3 (26.9)
Total	72.8 (22.7)	77.1 (24.5)	64.9 (30.9)	76.5 (13.3)

**Table 3 ijerph-20-03180-t003:** Outcomes of breathing and mindfulness at baseline (pre) and after 8 weeks of practice (post).

	SPB		M		SPB + M	
	pre	post	pre	post	pre	post
Average breathing rate during practice	6.0 (0)	6.0 (0.2)	11.9 (3.9)	11.7 (5.1)	6.0 (0)	6.0 (0)
Langer Mindfulness Scale (LMS)						
Total	100.2 (11.5)	99.6 (11.8)	106.5 (13)	111.7 (6.8)	107.7 (10.4)	109.7 (11.6)
Flexibility	19.2 (1.6)	18.8 (2.5)	21 (3.7)	23.2 (2.2)	19.4 (3)	19.6 (2)
Novelty seeking	28.6 (8.7)	29.6 (6.3)	34.8 (5.5)	34 (3.7)	35.1 (2.7)	34.7 (3.6)
Novelty producing	26.6 (2.2)	26 (2.5)	25.3 (5.1)	27.3 (4.5)	26.3 (6.4)	28.4 (6.3)
Engagement	25.8 (3.8)	25.2 (4.9)	25.3 (4.8)	27.2 (2.9)	26.9 (2.7)	27 (2.7)
Five Facet Mindfulness Questionnaire (FFMQ)						
Total	129 (24.5)	123 (33.8)	120.8 (9.9)	122.7 (16.6)	137.9 (8.1)	138.4 (13.2)
Observe	25 (6.2)	24 (6.5)	19.8 (3.1)	22.8 (3.6)	25.9 (2.7)	26.9 (2.9)
Describe	29.6 (7.4)	27.2 (10.4)	26.7 (5.3)	26.7 (6.3)	29.6 (3.9)	29.7 (5)
Act with awareness	26.6 (7.1)	24.8 (9.6)	26.3 (3.2)	25.3 (4.2)	28.7 (4.8)	29.4 (5.3)
Non-judging of inner experience	24.8 (9.1)	23.4 (10.5)	27.3 (8.8)	25.3 (10.7)	29.7 (6.6)	29.7 (7.5)
Non-reactivity to inner experience	23 (4.9)	23.6 (6.1)	20.7 (1.5)	22.5 (2.9)	24 (3.1)	22.7 (4)

## Data Availability

Data are available at https://scholar.harvard.edu/mgoldstein/ (accessed on 5 February 2023).

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
