# Peer review of "Dismantling the Component-Specific Effects of Yogic Breathing: Feasibility of a Fully Remote Three-Arm RCT with Virtual Laboratory Visits and Wearable Physiology"

_ijerph, 2023, doi:10.3390/ijerph20043180_

Round 1
Reviewer 1 Report
The research design is appropriate, and the manuscript is well-written. However, I have several concerns, as follows.
1. This study aimed to evaluate the feasibility of a three-arm intervention trial. According to Orsmond and Cohn (2015), Feasibility studies focus on the process of developing and implementing an intervention and result in preliminary examination of participant responses to the intervention, whereas pilot studies more clearly focus on outcomes, rather than process, and include a more controlled valuation of participant responses to the intervention. The authors should distinguish between feasibility study and pilot study.
2. Line 68: The authors should provide citations for studies with remotely delivered mindfulness-based interventions.
3. Lines 158–163: Please describe in detail how the authors recorded daily logs. In other words, how did the authors identify the sleep and practice periods?
4. Lines 186–189: The authors should provide a citation for determining the feasibility threshold.
5. Lines 186–189: Why was the feasibility threshold for overall daily practice adherence ratio only 50%, even though the other outcomes’ thresholds were 70%?
6. Please add the descriptions of the baseline and mindfulness outcomes in the Methods section.
7. Lines 203–204: The authors only reported descriptive statistics and did not test intervention efficacy or group-wise differences. However, they mentioned the differences in outcomes between and within groups in the Results and Discussion sections, such as “progressive decline in these group-averaged rates over the 8 weeks (Line 224–225).” Furthermore, the authors emphasized understanding unique effects of SPB and M versus their combination. Therefore, please consider performing the statistical comparative analyses and reporting effect sizes.
8. Supplementary Figures 1–5: Please add a color scale that shows power spectral density.
9. All Tables and Figures: All abbreviations should be spelled out in the footnotes.
10. The screening-to-randomization ratio was low (20%). Please discuss the limitations of this study, given the characteristics of the subjects to whom the remotely delivered mindfulness-based interventions should be provided.
Reviewer 2 Report
In this study, the author test the feasibility of a three-arm intervention trial, consisting of 20-min daily practice for 8 weeks with virtual laboratory visit and wearable devices. The authors did an excellent job of explaining the experimental design and method. Here are a few suggestions that may make your paper more useful to readers:
1) Line 57, Is that correct –(Kabat-Zinn[22])?
2) Would you please provide more details about the participants? Are they college-going/professional? Did you exclude/include pregnant women?
3) Line no-324-343 You reported multiple studies that have tested the feasibility in different populations. Would you please add more information if any of these studies reported any improvement in stress management/ psychological well-being?
Reviewer 3 Report
The study by Ma et al. is remarkably well written and deserves to be read carefully. Furthermore, it is to be appreciated that the authors describe in great detail the procedures followed and honestly acknowledge the limitations of the current pilot design. In short, this study is a methodological validation study in which the application of wearables for a telehealth intervention is highlighted. However, I do not see the direct application of analyzing its feasibility for the interventions and variables described. I mean, beyond the interest I have taken in tracking respiration through HRV, the results shown are rather trivial.
Further comments:
1. Very little information is displayed in the introduction about the effects of the three interventions on the tested variables. The authors state there is a growing evidence in the field, but no previous studies are presented thereafter.
2. The same could be said for existing evidence on wearable-based telehealth interventions. The authors should justify their decisions based more on previous studies.
3. Do you have any clues as to why there was lower adherence in the mindfulness group?
4. If the breathing rate in the mindfulness group varies from 4 to 17 bpm, the results may be compromised. It seems some participants used slow-paced breathing or they did not properly understand the intructions provided.
5. I miss in the discussion some reference to previous feasibility studies that justify the margins accepted as optimal by the authors. After all, this is the sole purpose of the study so it should be discussed at length.
Minor comments:
Lines 311-ff: The framework discussed here should first be introduced in the corresponding section.
Line 322: please put the 46 in parentheses
Round 2
Reviewer 1 Report
I would like to thank the authors for considering my comments and making the suggested edits. I think the revised manuscript is improved.
Reviewer 3 Report
I thank the authors for the work done to improve the original document. I am now satisfied with the changes made.